# Comparing and assessing four AI chatbots' competence in economics

**Patrik T. Hultberg**[1]*, **David Santandreu Calonge**[2], **Firuz Kamalov**[3], **Linda Smail**[4]

1 Department of Economics and Business, Kalamazoo College, Kalamazoo, Michigan, United States of America, 2 Department of Academic Development, Mohamed bin Zayed University of Artificial Intelligence, Abu Dhabi, United Arab Emirates, 3 School of Engineering, Applied Science and Technology, Canadian University Dubai, Dubai, United Arab Emirates, 4 College of Interdisciplinary Studies, Zayed University, Dubai, United Arab Emirates

* Patrik.Hultberg@kzoo.edu

**Data Availability Statement:** Data relevant to this study are available from GitHub at https://github.com/group-automorphism/tuce#readme.

**Funding:** The authors received no specific funding for this work.

## Abstract

Artificial Intelligence (AI) chatbots have emerged as powerful tools in modern academic endeavors, presenting both opportunities and challenges in the learning landscape. They can provide content information and analysis across most academic disciplines, but significant differences exist in terms of response accuracy for conclusions and explanations, as well as word counts. This study explores four distinct AI chatbots, GPT-3.5, GPT-4, Bard, and LLaMA 2, for accuracy of conclusions and quality of explanations in the context of university-level economics. Leveraging Bloom's taxonomy of cognitive learning complexity as a guiding framework, the study confronts the four AI chatbots with a standard test for university-level understanding of economics, as well as more advanced economics problems. The null hypothesis that all AI chatbots perform equally well on prompts that explore understanding of economics is rejected. The results are that significant differences are observed across the four AI chatbots, and these differences are exacerbated as the complexity of the economics-related prompts increased. These findings are relevant to both students and educators; students can choose the most appropriate chatbots to better understand economics concepts and thought processes, while educators can design their instruction and assessment while recognizing the support and resources students have access to through AI chatbot platforms.

## Introduction

In the dynamic landscape of education, the integration of technology has become pivotal in facilitating effective learning experiences [1]. The field of economics, known for its intricate concepts and theories, often poses challenges for students to grasp and internalize. Traditional methods of teaching, though valuable, may fall short in catering to individualized learning needs and fostering active engagement. Recognizing this gap, educators are exploring innovative tools like chatbots to enhance students' learning journey. These chatbots, driven by advancements in natural language processing (NLP) and machine learning (ML), have the potential to revolutionize how economics students interact with course content.

**Competing interests:** The authors have declared that no competing interests exist.

## Contributions of the paper

This article makes several contributions to the current literature:

1. *Improved Engagement*: The article highlights how chatbots can enhance student engagement by providing an interactive and accessible learning experience. Chatbots are today widely available, allowing students to access instant help and information at their convenience. This accessibility helps bridge the gap between classroom learning with an instructor and real-world applications, making economics concepts less abstract, more tangible, and relevant.

2. *Personalized Learning*: The article emphasizes the ability of AI chatbots to offer personalized learning experiences. By using AI algorithms, chatbots can tailor content, answers, and explanations to individual students' needs and learning preferences. This personalized approach ensures that students receive targeted support and explanations (and paraphrasing of the explanations with the *regenerate answer* option), which may make complex economic concepts more digestible.

3. *Instant Feedback and Assessment*: One significant contribution of the study is its exploration of how chatbots can provide detailed explanations on prompts that require higher levels of thinking (analyze/evaluate). This finding has not previously been described and stands out from the results reported in the published literature. Chatbots can also quiz students, provide feedback and explanations for incorrect answers, and track their progress. This immediate feedback loop helps students identify gaps of knowledge/areas of weakness and reinforces their understanding of economic principles.

4. *Real-World/Practical Applications*: This article discusses how chatbots can simulate and answer real-world economic scenarios (TUCE and advanced prompts), helping students think like economists. Students can thus engage with these chatbots to practice economic decision-making in a controlled environment. This practical application of theory helps students bridge the gap between classroom knowledge and real-world economic challenges.

5. *Overcoming Barriers to Learning*: This article addresses common barriers that economics students may face, such as understanding how various economic agents interact, or fear of asking questions in class, or limited access to resources in some contexts/countries. Chatbots provide a safe and non-judgmental space for students to seek clarification and assistance, helping them overcome these obstacles.

## Literature review

**Emergence of chatbots for learning and support.** In recent years, the landscape of higher education has been significantly transformed by technological advancements, with chatbots emerging as a prominent innovation that has revolutionized the way students engage with learning resources [2–5]. Chatbots, powered by artificial intelligence (AI), have become invaluable tools in the higher education sector, enhancing student experiences, improving administrative efficiency, and providing personalized learning support [2, 3, 6–14].

The integration of chatbots into higher education has been driven by several factors. First, the rise of online and remote learning has necessitated new methods of communication and support for students who may not have easy access to in-person assistance. Chatbots offer a convenient way for students to seek information and guidance in real time, regardless of their physical location [15]. This on-demand access to information contributes to a more inclusive learning environment and supports diverse student populations [16]. One of the primary

benefits of chatbots in higher education is their ability to provide personalized learning experiences [2]. Through AI algorithms, chatbots can analyze student behaviors, preferences, and performance data to deliver tailored recommendations for learning resources, study strategies, and even course selections. This personalized approach fosters self-directed learning and helps students navigate the complexities of their academic journey more effectively.

Furthermore, chatbots have been instrumental in streamlining administrative processes within higher education institutions. Tasks such as course registration, scheduling, and accessing campus services can be automated through chatbot interactions. This not only reduces administrative burdens but also empowers students to manage their academic affairs independently, leaving university staff with more time to focus on higher-value tasks. Language barriers can also be overcome with the help of chatbots that support multilingual communication. International students, in particular, benefit from chatbots that can provide information and guidance in their native languages, ensuring they are well informed and comfortable in their new academic environment.

In summary, the emergence of chatbots in higher education marks a pivotal development in the way students learn and interact with their academic institutions. These AI-powered tools enhance personalization, accessibility, and administrative efficiency, contributing to an enriched learning experience for students. As technology continues to evolve, the potential for chatbots to reshape education for the better remains an exciting prospect, offering endless opportunities for innovation and improvement in higher education.

**Benchmarking chatbot performance.** Large language models (LLMs) have attracted a great deal of interest from the research community. Many studies have attempted to compare the performance of LLMs in several subject matters. A large study that evaluated LLMs showed that GPT-4 can solve novel and difficult tasks that span mathematics, coding, vision, medicine, law, psychology and more, without needing any special prompting [17]. Moreover, in all these tasks, GPT-4's performance is strikingly close to human-level performance, and often vastly surpasses prior models such as ChatGPT. The performance of generations of GPT, GPT-3, GPT-3.5, and GPT-4, in programming courses was carried out by Savelka et al. (2023) who found a significant rate of improvement which strongly suggests their potential to handle almost any type of assessment widely used in higher education programming courses [18].

Additionally, it was shown by Katz et al. (2023) that GPT-4 performs well on law-related questions [19]. In particular, the authors found that on the Multistate Bar Examination, GPT-4 significantly outperformed both human test-takers and prior models, demonstrating a 26% increase over ChatGPT and beating humans in five of seven subject areas. LLMs have also been tested extensively in the medical field. Nori et al. (2023) analyzed the performance of GPT-4 on medical challenge problems and found that GPT-4 exceeds the passing mark on the United States Medical Licensing Examination (USMLE) by 20 points even with no specialized prompt crafting [20]. GPT-4 not only outperformed its predecessor, GPT-3.5, but also surpassed models that were explicitly adjusted for medical information, such as Med-PaLM. The three major chatbots, GPT-4, GPT-3.5, and Bard, were compared in the context of neurosurgery examination [21]. The results showed that GPT-4 outperformed GPT-3.5, and Bard by scoring 82.6%, 62.4%, and 44.2%, respectively. The performance of GPT-4 on non-English problems was tested by Takagi et al. (2023) who applied GPT-4 on the Japanese Medical Licensing Examination (NMLE). The results showed that GPT-4 outperformed GPT-3.5, particularly for general, clinical, and clinical sentence questions. It also attained the exam passing mark [22]. Similar results were found by Kasai et al. (2023) [23].

While most of the studies have extolled the performance of GPT-4, not all the studies have made positive conclusions regarding the performance of LLMs. Espejel et al. (2023) found for

instance that both GPT-3.5 and GPT-4 exhibited limited performance in Inductive, Mathematical, and Multi-hop Reasoning Tasks [24].

**Chatbots in economics education.**   The research into the use of AI chatbots in economics education has been limited. One of the earliest studies on the effectiveness of chatbots in economics courses was conducted by Terwiesch (2022) who found that GPT-3 was capable of excellent achievement on the final examination in the Operations Management course at Wharton's MBA program, albeit with some surprising mistakes in basic calculations [25]. The implementation of chatbots in economics teaching was considered by Cowen and Tabarrok (2023) who tried to guide instructors to the best practice conventions [26]. A guide to the use of AI chatbots for management scholars was proposed by Rana (2023) [27]. Cribben and Zeinali (2023) discussed the benefits and limitations of ChatGPT in business education and research, with a particular focus on the areas of management science, operations management and data analytics [28]. A recent study by Geerling et al. (2023) explored the ability of GPT-3.5 to provide correct answers to the Test of Understanding in College Economics (TUCE) multiple-choice questions and compared its ability to that of college-level students in the United States, they found that GPT-3.5 was able to outperform students of introductory economics [29].

Chatbot-assisted learning offers several advantages for economics students [30, 31]. Zhang et al. (2023) argued for instance that chatbots can provide "support through presenting knowledge, facilitating practices, supervising and guiding learning activities, and. . .emotional support" (p.1) [30]. Firstly, chatbots can provide immediate access to help, information, and answers to economic issues facing the national or world economy, allowing students to clarify doubts and deepen their understanding in real time, fostering a more engaging and dynamic learning experience. Secondly, chatbots can adapt to individual learning paces and preferences [32], offering personalized content and recommendations on topics such as inflation, scarcity, or market failure, which enhances engagement, understanding, and retention. Moreover, they are available 24/7, accommodating flexible study schedules and catering to the needs of a global and connected student population. Additionally, chatbots can simulate real-world economic scenarios with realistic projections and offer practical problem-solving exercises, preparing students for the challenges they may encounter in their careers. Overall, chatbot-assisted economics learning can be a valuable tool to empower students to excel in their studies while promoting self-directed learning and critical thinking skills.

**Ethical considerations and data privacy.**   Ethical considerations and user's data privacy are integral to the successful integration of chatbots in higher education [3, 33, 34]. Institutions must ensure that data collected by chatbots is handled responsibly and transparently, with proper safeguards in place to protect student privacy. Additionally, efforts should be made to strike a balance between AI-driven interactions and maintaining opportunities for genuine human engagement [35], which remains essential for certain complex situations.

## Methods

The study contained two phases, in the first phase ("baseline") the responses of four AI chatbots were compared across their ability to generate accurate conclusions and explanations to 60 questions from the Test of Understanding in College Economics (TUCE). These multiple-choice questions mainly tested the knowledge, understanding, and ability to apply concepts learning in a standard principles of economics sequence, including both microeconomics (30 questions) and macroeconomics (30 questions). This initial step of the analysis was designed to reproduce and extend findings from Geerling et al. (2023) that explored the ability of Open-AI's ChatGPT (3.5) ability to provide correct answers to the TUCE multiple-choice questions

and compared its ability to that of college-level students in the United States [29]. The current study's difference from Geerling et al. (2023) included the use of TUCE version III, as opposed to TUCE version IV, including three additional AI chatbots, GPT-4, Google's Bard, and Meta's LLaMA 2, and a careful analysis of the explanations provided by the chatbots. To reduce variability in response assessment, all explanations were scored using a defined rubric.

To gauge the level of agreement and inter-rater reliability in the analysis of the responses generated by the four chatbots, Krippendorff's alpha coefficient was employed, as introduced by Krippendorff in 2011, with a result of 0.87 [36]. Krippendorff's alpha is a versatile measure that can be applied to assess inter-rater reliability in situations where there are multiple raters and variables, and it accommodates both nominal and ordinal data. Given the complexity of evaluating the subtleties and nuanced responses to economics prompts (TUCE and advanced prompts) generated by the four AI chatbots, Krippendorff's alpha offered a robust and adaptable method to assess the degree of agreement among raters and ensure reliable and consistent findings.

A rubric is a scoring tool that presents the different expectations for an assignment [37, 38]. This study's proposed rubric divided each assignment into two parts and assigned a description of how each part was evaluated. For this study the main goals were to (1) establish whether the conclusion, the final answer, was correct, and (2) establish whether the explanation provided used appropriate concepts and whether those concepts were used appropriately. Both criteria are important for students' ability to use and trust the responses provided by the generative language model, as well as for economics educators to recommend or use an AI tool in their teaching. Of course, if the response given is incorrect, then students who rely on the LLM will either provide an incorrect answer or learn the wrong lesson. This is likely to be true even if the LLM provides an explanation that an expert could recognize as correct and therefore could use to override the incorrect final conclusion. Alternatively, if the conclusion is correct but accompanied by an incorrect or incomplete explanation, then students are most likely learning less (or absorbing incorrect lessons) and may give the wrong answer in a test/quiz. The rubric used to score responses to the TUCE III experiment is shown in Table 1 below.

**Table 1. Rubric used to score AI chatbot responses to TUCE III.**

| Score | Accuracy of Conclusion | Quality of Explanation Provided |
|---|---|---|
| 1 | Principal conclusion is incorrect, or response does not include a principal conclusion. | • Explanation uses inappropriate concepts.<br>• Explanation uses relevant concepts inappropriately.<br>• No explanation is provided. |
| 2 | Principal conclusion is incorrect, or response does not include a principal conclusion. | • Explanation uses relevant concepts appropriately but arrives at the incorrect conclusion.<br>• Explanation uses relevant concepts appropriately but arrives at an ambiguous conclusion. |
| 3 | Principal conclusion is correct | • Explanation uses inappropriate concepts.<br>• Explanation uses relevant concepts inappropriately. |
| 4 | Principal conclusion is correct | • Explanation uses relevant concepts, but concepts are used inappropriately.<br>• Explanation and/or examples are not appropriate for a college-level student.<br>• Explanation is relevant and appropriate, but argument is not concluded ("stops in the middle"). |
| 5 | Principal conclusion is correct | • Explanation uses relevant concepts appropriately and arrives at the correct conclusion. In addition, the response uses explanation and/or examples that are appropriate for a college-level student. |

The second phase ("extension") of the study investigated whether the results were different when the four AI chatbots were prompted with more advanced economics concepts and problems; that is, are AI chatbots equally capable of responding to the more complex cognitive learning tasks of analysis and evaluation? Ten advanced economics prompts were designed by the authors across three microeconomics topics and two macroeconomics topics: for each topic one prompt targeted analysis (often mathematical analysis) and one prompt asked a more open-ended, evaluative question (see S1 Appendix for the included prompts). Due to the qualitatively different task, a different rubric was used to score the accuracy and quality of the AI chatbots' responses, as shown in Table 2 below.

## Results

### Phase 1: Baseline

Geerling et al. (2023) examined whether ChatGPT (GPT-3.5) was able to correctly answer questions on a test of college-level introductory economics [29]. Using the Test of Understanding in College Economics (TUCE), a standardized test of principles-level knowledge and analysis that is often given as a pre- and post-test to students of economics in the United States, Geerling et al. (2023) found that GPT-3.5 ranked in the 91st percentile for microeconomics understanding and the 99th percentile for macroeconomics understanding when compared to students who took the TUCE exam (version IV) at the end of their respective principles course. This translated into GPT-3.5 correctly answering 19 of 30 microeconomics questions and 26 of 30 macroeconomics questions on the standardized test. If the chatbot was compared to students who took TUCE as a pre-test, then GPT-3.5 scored at the 99th percentile for both microeconomics and macroeconomics.

As a baseline examination, the current study attempts to initially reproduce the results of Geerling et al. (2023) with two modifications. First, the current study uses TUCE version III

**Table 2. Rubric used to score AI chatbot responses to advanced prompts.**

| Score | Accuracy of Conclusion | Quality of Explanation Provided |
|---|---|---|
| 1 | Solution is incorrect | • Explanation uses inappropriate concepts.<br>• Explanation uses relevant concepts inappropriately.<br>• No part of the solution is correct.<br>• Fails to set up the problem correctly. |
| 2 | Solution is incorrect | • Explanation uses relevant concepts appropriately, but less than half of the question is correctly answered. Example: Correctly solves the first step (1–2) in a five-step solution.<br>• Provides relevant discussion for less than half of the relevant concepts and arrives at an incorrect or incomplete conclusion. |
| 3 | Solution is incorrect | • Explanation uses relevant concepts appropriately and more than half of the question is correctly answered. Example: Correctly solves the first three steps (3–4) in a five-step solution.<br>• Provides relevant discussion for more than half of the relevant concepts and arrives at an incorrect or incomplete conclusion. |
| 4 | Solution is correct. | • Explanation uses relevant concepts appropriately and all the entire question is correctly answered. Example: Correctly solves all the steps (5) in a five-step solution.<br>• Provides relevant discussion of "all" relevant concepts and arrives at a correct conclusion. |

[39] to broaden the set of questions used to evaluate the use of AI chatbots and, second, the study compares the results from using GPT-3.5 to using the upgraded version GPT-4, which is marketed as "great for tasks that require creativity and advanced reasoning" (GPT-4 is currently only available to Plus users, which require a subscription). The study also includes Bard, the AI chatbot offered by Google, and LLaMA 2, the AI chatbot introduced by Meta, the parent company of Facebook. TUCE version III is very similar to version IV used in Geerling et al. (2023), it is a standardized test of principles-level knowledge and analysis often given to students of economics in the United States. Both versions of the test allow instructors across different institutions to compare their students' performance with those of post-secondary students across the United States [39–41]. TUCE III also contain 60 multiple-choice questions, split across microeconomics and macroeconomics, and is designed so that an average student after completing a semester-long principles of economics course score around 40–50% of the questions correctly. That is, the test is challenging for most students. In particular, for TUCE III, students taking the test of microeconomics averaged 10.71 (35.7%) in the pre-test and 15.36 (51.2%) in the post-test. For macroeconomics, the results were 9.18 (30.6%) and 14.31 (47.7%) on the pre- and post-tests, respectively. Figs 1 and 2 show the distribution of the scores for the two tests. In addition, Table 3 summarizes the results of students on TUCE III.

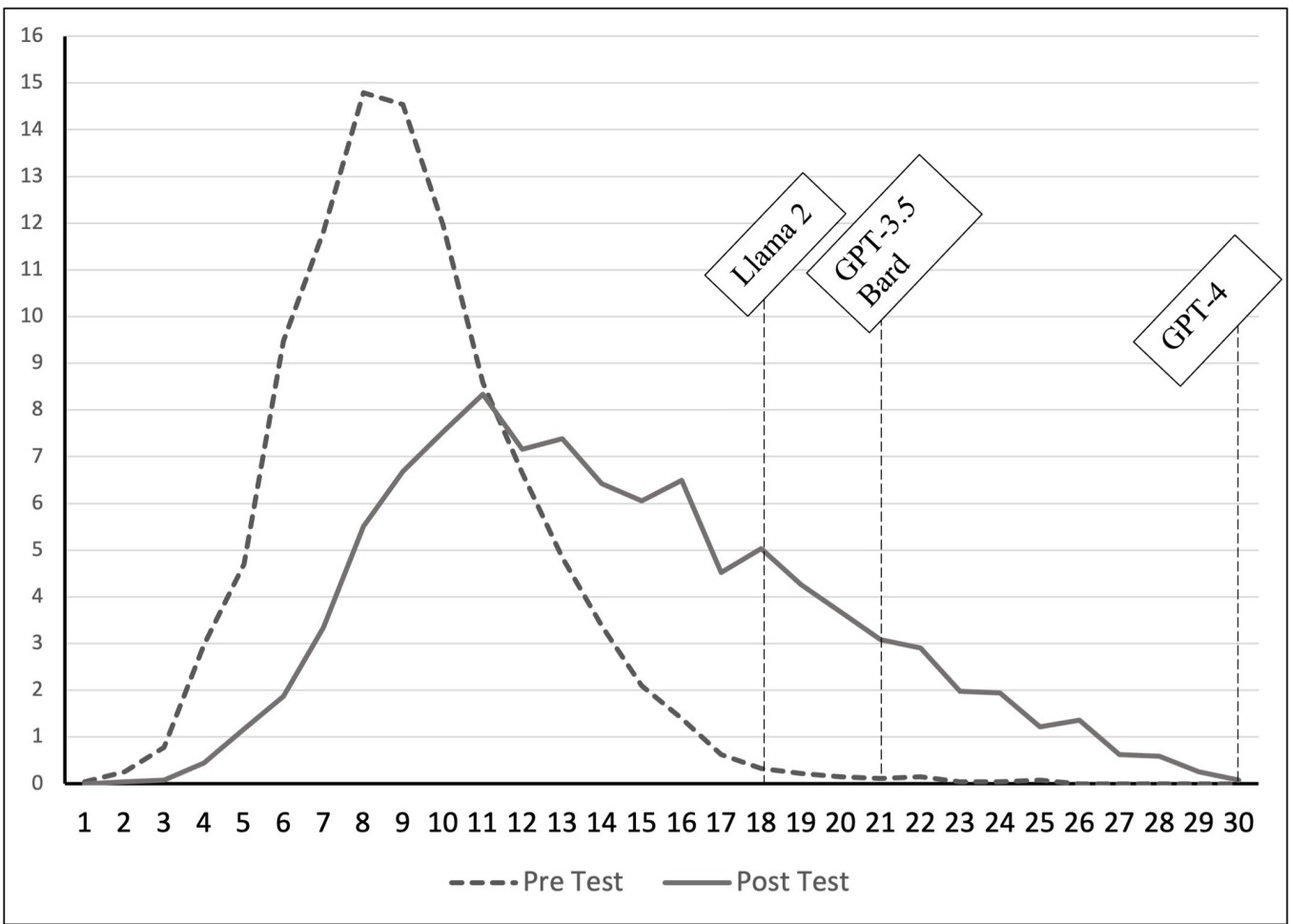

**Fig 1. Results on TUCE III microeconomics.**

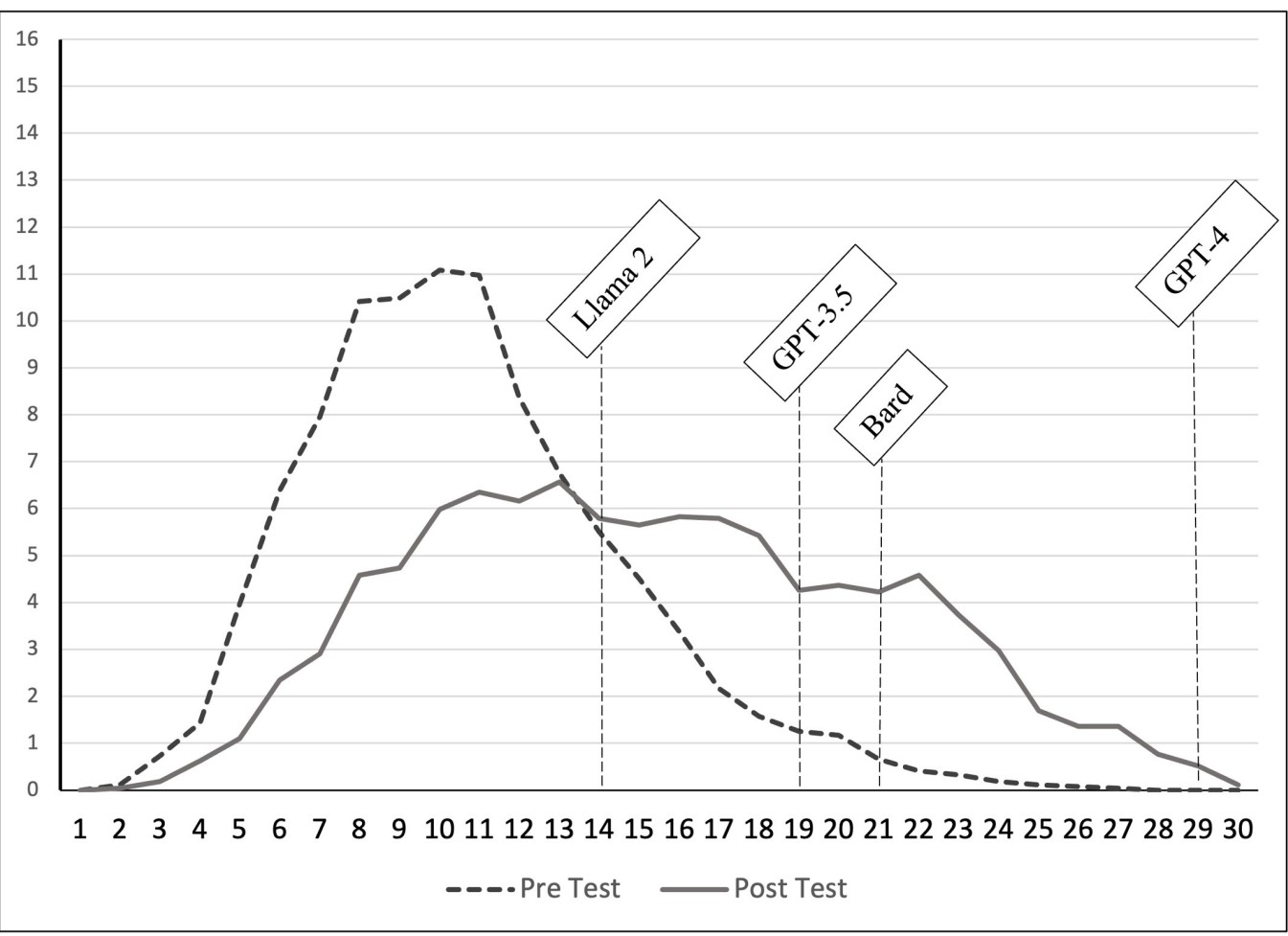

**Fig 2. Results on TUCE III macroeconomics.**

Applying GPT-3.5 to the previous version of TUCE yielded qualitatively similar results as described in Geerling et al. (2023); GPT-3.5 answered 21 of 30 questions correctly on the microeconomics test (2 more than its result on TUCE version IV) and 19 of 30 questions correctly on the macroeconomics text (7 fewer than its result on TUCE version IV). These results would have placed the chatbot at the 99th percentile among all students taking TUCE III as a pre-test. However, GPT-3.5 would only have scored at the 98th and 80th percentile for students who have completed the principles of economics course (post-test), respectively.

OpenAI describes GPT-4 as an enhanced version of GPT-3.5 and has lauded its capabilities. To test this assertion, the current research hypothesizes that GPT-4 performs significantly better than GPT-3.5, Bard, and Llama 2 in the context of undergraduate economics. To test this hypothesis, each model was evaluated on TUCE III, as well as more advanced prompts in economics. The null hypothesis was thus that each AI chatbot would score the same across the TUCE. The results, as shown in Tables 4–6, reject the null hypothesis and indicate that the difference between GPT-4 and the other algorithms is statistically significant in the context of undergraduate economics. In particular, extending the study to include additional AI chatbots yielded the following results: GPT-4 scored 30 of 30 on the microeconomics section and 29 of

**Table 3. Results for students on TUCE version III.**

| | Students TUCE III | | | |
|---|---|---|---|---|
| | Microeconomics (2,726 students) | | Macroeconomics (2,724 students) | |
| | PRE TEST | POST TEST | PRE TEST | POST TEST |
| Mean Score | 10.71 | 15.36 | 9.18 | 14.31 |
| Std. Deviation | 3.45 | 5.40 | 3.05 | 5.24 |

Results from Saunders (1991).

30 on the macroeconomics section; Bard scored 21 of 30 and 21 of 30, respectively; LLaMA 2 scored 18 of 30 and 14 of 30, respectively. Table 4 summarizes all the results and provides the percentile when compared to all students taking TUCE III. For comparison, the results from Geerling et al. (2023) using TUCE IV have also been included.

Comparing the four AI chatbots' performance across the TUCE III, it appears as if GPT-4 is far superior to the other three chatbots when answering multiple-choice questions targeting knowledge, understanding, and basic applications of economics concepts. This is true whether questions relate to microeconomics or macroeconomics. The results also indicate that, at this moment, Meta's LLaMA 2 is the least capable of correctly answering questions about basic economics. Finally, the free version of ChatGPT (3.5) and Google's Bard seem similar in their ability to tackle economics concepts. For a graphical depiction of the AI chatbots' performance, Figs 1 and 2 include their results along with the student distribution. Based on these preliminary results, students who wish to use an AI chatbot to answer multiple-choice economics questions are best served by using GPT-4, whether the task is a formative or summative assessment. Economics educators need to be aware of the power of AI chatbots in general, and GPT-4 in particular, when assessing students remotely.

Economics students and educators are of course not only concerned about finding the correct answer to a multiple-choice question, but they are also interested in the accompanying explanation. In fact, if the goal is to learn (or assess learning) it is often more important to assess the quality of the explanation provided along with the final conclusion. To explore this question, the current study went beyond Geerling et al. (2023) to also assess the quality of the AI chatbots' responses [29]. Using the rubric presented in Table 1, a subjective, but careful, assessment of all 240 responses was conducted. The combined results for both the microeconomics and macroeconomics parts of TUCE III are presented in Fig 3.

As shown by the histogram in Fig 3, the GPT-4's responses that accompanied the correct answers were of consistently high quality (rated 5 based on the rubric from Table 1). In fact, the only incorrect response was accompanied by a good and comprehensive explanation, which apparently prevented the chatbot from giving a final principal conclusion. The quality

**Table 4. Results across AI chatbots and compared to students on TUCE III.**

| | GPT-3.5 | | GPT-4 | | Bard | | LLaMA 2 | | GPT-3.5 (TUCE IV) | |
|---|---|---|---|---|---|---|---|---|---|---|
| Microeconomics | 21/30 | Pre: 98[th] | 30/30 | Pre 99[th] | 21/30 | Pre 98[th] | 18/30 | Pre 94[th] | 19/30 | Pre 99[th] |
| | | Post: 80[th] | | Post 99[th] | | Post 80[th] | | Post 67[th] | | Post 91[th] |
| Macroeconomics | 19/30 | Pre 99[th] | 29/30 | Pre 99[th] | 21/30 | Pre 99[th] | 14/30 | Pre 93[th] | 26/30 | Pre 99[th] |
| | | Post 80[th] | | Post 99[th] | | Post 92[th] | | Post 52[th] | | Post 99[th] |

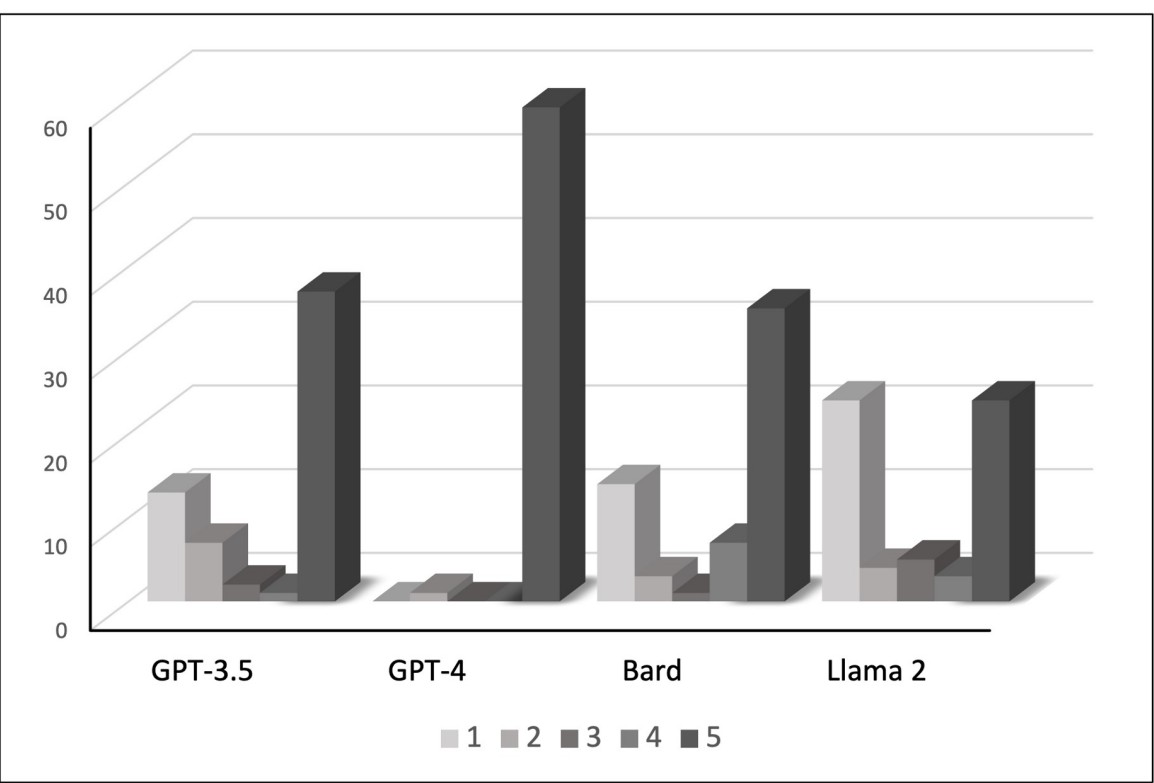

**Fig 3. Assessment of quality of responses to TUCE III.**

of responses given by GPT-3.5 and Bard were again similar, with GPT-3.5 performing slightly better. Once more, LLaMA 2 struggled in providing quality explanations in response to the economics questions. Another issue with LLaMA 2 was that the response word count was limited to approximately 200 words by the official application used, this meant that several responses were incomplete. However, for TUCE III this did not materially affect the ability to evaluate the responses (it was more of an issue for the advanced prompts discussed below).

While it is evident from Table 4 that there is a difference between the performances of the chatbots on TUCE III, the difference may not be statistically significant given the relatively small sample size. To ascertain the significance of the difference in accuracy, a two-tailed, two-sample test of proportions was conducted. The null hypothesis in the test is that the proportion of correct answers is the same for the pair of chatbots under consideration; that is, that the two chatbots have the same accuracy. The z- and p-values of the test for microeconomics and macroeconomics are presented in Table 5. The results for microeconomics show that GPT-4 achieves a statistically significant performance over the rest of the chatbots, while the difference between the GPT-3.5, Bard, and LLaMA 2 is not statistically significant. The results for macroeconomics match the results for microeconomics and show a significant difference between the accuracy of GPT-4 and the other chatbots.

Similarly, Fig 3 indicates that there is a difference in the quality of the explanations between the chatbots. To determine if the difference is statistically significant, we performed a two-tailed, two-sample (unpooled) t-test. The null hypothesis in the test is that the average quality of the explanations is the same for the pair of chatbots under consideration. The t- and p-values of the test for microeconomics and macroeconomics are presented in Table 6. For

**Table 5. Pairwise comparison of accuracy, TUCE III.**

| Pair | Microeconomics | | Macroeconomics | |
|---|---|---|---|---|
| | z-value | p-value (two-tailed) | z-value | p-value (two-tailed) |
| GPT-3.5 vs GPT-4 | 3.25 | ~ 0 | 3.23 | ~ 0 |
| GPT-3.5 vs Bard | 0 | 1.0 | 0.55 | 0.58 |
| GPT-3.5 vs LLaMA | 0.81 | 0.42 | 1.30 | 0.20 |
| GPT-4 vs Bard | 3.25 | ~ 0 | 2.77 | ~ 0 |
| GPT-4 vs LLaMA | 3.87 | ~ 0 | 4.30 | ~ 0 |
| Bard vs LLaMA | 0.81 | 0.42 | 1.83 | 0.06 |

microeconomics, the results show a significant pairwise difference between the chatbots *except* for GPT-3.5 vs LLaMA 2. For macroeconomics, the results presented show that there is a significant pairwise difference between all the chatbots in terms of the quality of the provided explanations. Finally, for the advanced prompts (higher level of cognitive complexity) the pairwise comparison shows that the chatbots differ significantly from each other, except for Bard versus LLaMA 2.

Alongside the significance tests, we computed 95%-confidence intervals for the differences in proportions of correct answers (score 5). This gives a range in which the true difference likely falls and offers a sense of the magnitude and uncertainty of the differences. The results are shown in Table 7.

The true percentage of correct solutions for GPT-4 is estimated to be between 94% and 103%, although a percentage greater than 100% is nonsensical. The upper bound simply indicates a very high level of confidence that the GPT-4 response will provide the correct solution on a test of basic economics. When comparing confidence intervals across the four AI chatbots, GPT-4 stands out as the most reliable and best-performing chatbot at this moment in time. Its confidence interval does not overlap with any other chatbot, indicating a significant difference in performance and supporting the results from Tables 5 and 6. Second, GPT-3.5 and Bard have overlapping confidence intervals. This overlap suggests a lack of significant differences in their performances, and they can both be considered as middle performers. Among the two, GPT-3.5 has a slightly higher lower bound, suggesting that it might be a bit more consistent than Bard. Lastly, LLaMA 2 has the lowest upper bound among all chatbots and its interval suggests that it is the weakest chatbot in terms of performance when applied to economics concepts.

## Phase 2: Advanced prompts

Being a multiple-choice test given to introductory students of economics, TUCE focuses on the lower levels of cognitive learning complexity as defined by Bloom's taxonomy; that is, the

**Table 6. Pairwise comparison of quality of explanation, TUCE III and advanced prompts.**

| Pair | Microeconomics | | Macroeconomics | | Advanced Prompts | |
|---|---|---|---|---|---|---|
| | t-value | p-value | t-value | p-value | t-value | p-value |
| GPT-3.5 vs GPT-4 | -27.747 | < 0.0001 | -5.846 | < 0.0001 | -1.950 | ~0.07 |
| GPT-3.5 vs Bard | -4.734 | < 0.0001 | -3.062 | < 0.01 | 2.686 | < 0.05 |
| GPT-3.5 vs Llama 2 | 1.546 | ~0.13 | 2.796 | < 0.01 | 3.717 | < 0.01 |
| GPT-4 vs Bard | 24.789 | < 0.0001 | 2.819 | < 0.01 | 5.264 | < 0.0001 |
| GPT-4 vs Llama 2 | 47.770 | < 0.0001 | 9.371 | < 0.0001 | 10.699 | < 0.0001 |
| Bard vs Llama 2 | 5.819 | < 0.0001 | 6.738 | < 0.0001 | 1.708 | ~0.11 |

**Table 7. 95% Confidence intervals for the mean scores of each chatbot.**

| 95% CI TUCE III (score 5) | GPT-3.5 | GPT-4 | Bard | LLaMA 2 |
|---|---|---|---|---|
| Lower bound | 44% | 94% | 41% | 22% |
| Upper Bound | 79% | 103% | 76% | 58% |

questions require students to remember economics concepts, understand those concepts sufficiently to use the material to make an inference, and to apply economic concepts to solve a basic economic problem from a standard principles of economics course [42]. However, many economics educators ask their students questions that require higher levels of thinking and higher cognitive learning complexity. To explore the AI chatbots' ability to correctly respond to prompts that ask the student (or, in this case, the AI chatbot) to either analyze or evaluate a situation, a similar assessment was performed based on ten prompts (six related to microeconomics and four related to macroeconomics) created by the authors of this study. The actual questions asked are reported in S1 Appendix. Each response to these ten prompts by the four AI chatbots was assessed using the rubric presented in Table 2. The results are shown in Fig 4.

Although the results are superficially similar to the findings from the TUCE III analysis, significant differences emerge. First, GPT-4 is still the AI chatbot that performs the best when confronted with economic issues/questions. Although GPT-4 was not able to correctly answer all the prompts, the explanations provided were appropriate/accurate and concepts were

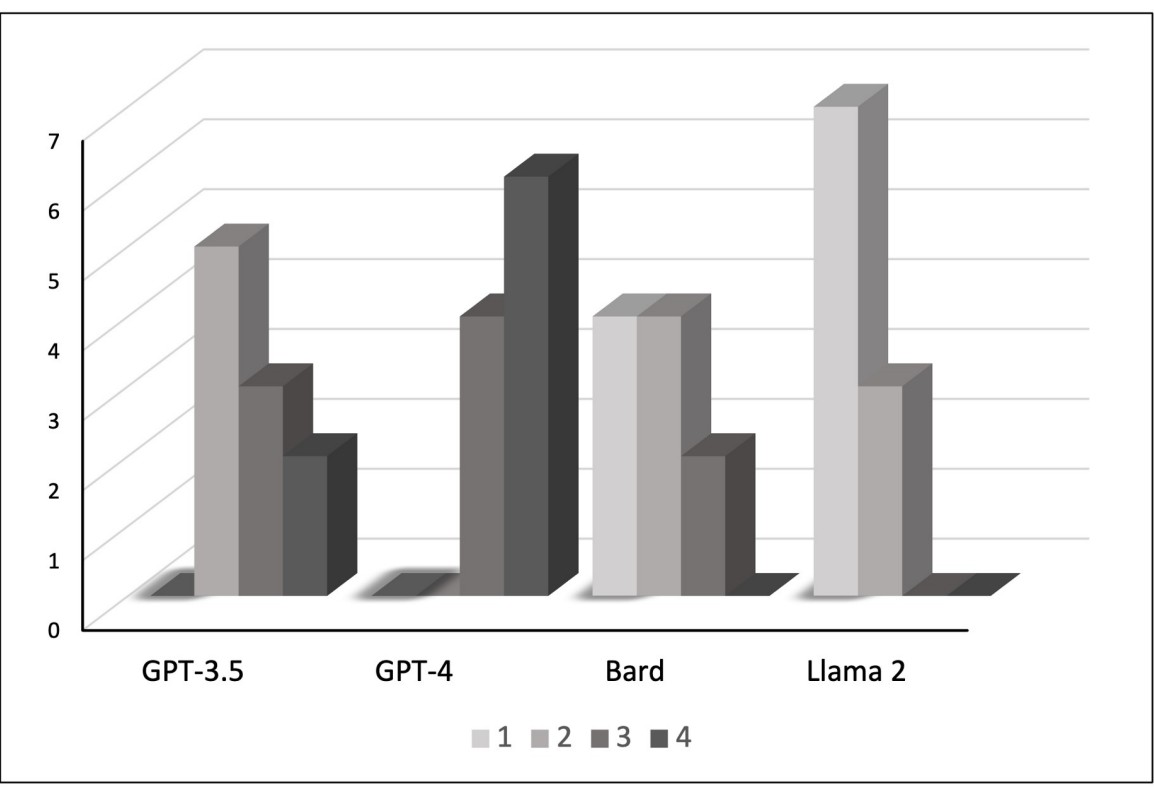

**Fig 4. Scored responses to ten advanced economics prompts.**

mostly used appropriately, nevertheless an incorrect final conclusion was reached 40 percent of the time. Contrast this to LLaMA 2 and Bard, both of which never provided the correct analysis or an appropriate evaluation of the scenarios. However, even among these lower performers a distinction can be made, Bard mostly used concepts that were relevant and potentially appropriate. The same cannot be said for LLaMA 2. Finally, GPT-3.5 produced intermediate performance. Although a correct answer was only given 20 percent of the time, all responses applied relevant concepts in their analysis—although 80 percent of the time the response produced an incorrect final conclusion.

Based on these results, students who wish to use an AI chatbot for higher-level cognitive tasks should exercise caution. Except for GPT-4, the effort is likely to produce an incorrect answer and an incomplete explanation. For economic educators this might be "a feature, not a bug." That is, at this point it might still be possible to design a complex economics problem that cannot effectively be solved by an AI chatbot.

### Word count analysis

Although the *accuracy* of the AI chatbot's response is most important, another notable difference between the LLMs is the wordcount of their responses and the level of verbosity displayed. To explore these apparent differences, average wordcounts and standard deviations were calculated. Table 8 displays the results, which show that for TUCE III, GPT-3.5 offered the fewest words in its response, followed by GPT-4, and then Bard. The analysis does not include LLaMA 2 as its responses were often cut short by the application used. Also, a few times (3) the response of Bard was "not programmed to assist with that." These occasions were excluded from the analysis. Interestingly, although the sample size was very limited, the responses to the advanced prompts showed a reversal of these results as in these cases the response of Bard was the least verbose, followed by GPT-4, and then GPT-3.5.

To verify the results in Table 8, two-tailed, two-sample t-tests of means were conducted. The tests show that across all the responses, the differences between GPT-3.5 and GPT-4 were not statistically significant. However, their responses were significantly different from those of Google's Bard chatbot, except for the case of the more advanced prompts. See Table 9.

## Conclusions and implications for practice

### Summary of findings

The current study, especially the initial baseline investigation, established that there exists an AI chatbot (GPT-4) that easily outperforms even advanced students on concepts from a principles of economics sequence. At the same time, there is a significant difference in performance across different AI chatbots as GPT-3.5, Bard, and LLaMA 2 all performed worse than GPT-4 on tasks related to lower cognitive learning complexity, as defined by Bloom's taxonomy. However, even these AI chatbots perform better than an average principles of economics students, as well as an average student that has completed the principles of

**Table 8. Word count analysis.**

|  |  | GPT-3.5 | GPT-4 | Bard |
|---|---|---|---|---|
| TUCE III | Mean | 140.3 | 170.9 | 273.3 |
|  | Std. Dev. | 103.2 | 96.6 | 62.0 |
| Advanced Prompts | Mean | 398.0 | 351.1 | 320.3 |
|  | Std. Dev. | 74.8 | 90.2 | 43.8 |

Table 9. Pairwise comparison of word counts, TUCE III and advanced prompts.

| Pair | TUCE Economics | TUCE Microeconomics | TUCE Macroeconomics | Advanced Prompts |
|---|---|---|---|---|
| | p-value | p-value | p-value | p-value |
| GPT-3.5 vs GPT-4 | < 0.01 [1] | 0.072 [1] | < 0.05 [2] | 0.222 [1] |
| GPT-3.5 vs Bard | < 0.0001 [1] | < 0.0001 [2] | < 0.0001 [2] | 0.011 [1] |
| GPT-4 vs Bard | < 0.0001 [2] | < 0.0001 [1] | < 0.0001 [2] | 0.349 [2] |

[1] F-test indicated equal variance, so t-test assumed equal variance.

[2] F-test indicated unequal variance, so t-test assumed unequal variance.

economics sequence. Thus, students can use these platforms for both learning and assessment purposes. Similarly, economics educators need to be acutely aware of the ability of AI chatbots to answer multiple-choice questions and provide basic explanations to problems that target students' ability to remember, understand, and apply concepts. To illustrate this, Table 10 provides one example of an explanation provided by GPT-4 to a multiple-choice question.

In the second phase of the study, several prompts that venture to address higher levels of cognitive complexity were explored. It was shown that the GPT-4 chatbot once again outperformed its main competitors in analytical and evaluative tasks. Despite being much less proficient in both ability to arrive at the correct final conclusion and provide accurate and complete explanations on these higher thinking tasks, GPT-4 was able to mostly analyze the questions. Its predecessor, GPT-3.5, struggled to provide a correct final answer and often used relevant concepts inappropriately or inaccurately. Bard and LLaMA 2 did even worse on these analytical tasks and their use is not recommended for students of economics. To give a sense of the GPT-4 performance, a response to a microeconomics prompt is shown in Table 11.

Table 10. One GPT-4 response to a multiple-choice question.

**QUESTION**
"The effect of an excise tax on the products of pollution-producing industries will be a cut back in production. If the tax was levied directly on the amount of pollution generated, however, the long-run cutbacks in production would be much smaller." This statement is most likely to be:

1. False, provided the amount of the taxes on products and pollution is equal.

2. True, because firms would have a greater incentive to adopt new technology that causes less pollution.

3. False, because most firms would rather pay the tax than cut back production.

4. True, because most taxes levied on pollution affect the demand curve; taxes on products affect the supply curve.

**GPT-4 RESPONSE**
The statement posits that if an excise tax is levied on the products of pollution-producing industries, it will lead to a cutback in production. If the tax, however, is levied directly on the amount of pollution generated, then the long-run cutbacks in production would be smaller.

The rationale behind this is that taxing the pollution directly gives firms an incentive to reduce their pollution levels without necessarily reducing their output. They can achieve reduced pollution levels by adopting cleaner technologies or more efficient production methods.

On the other hand, if the tax is levied on the products themselves, it raises the cost of producing each unit, which could lead to a reduction in output. Firms might not have as direct an incentive to just reduce their pollution–they might also cut back on production to reduce the tax they owe.

So, the correct statement is: True, because firms would have a greater incentive to adopt new technology that causes less pollution.

**Table 11. One GPT-4 response to an advanced prompt for microeconomics.**

**QUESTION**

Suppose a perfectly competitive firm produce candy bars according to the Cobb-Douglas production function, $Q = AK^{0.5} L^{0.5}$, where $A$ is 50. Suppose that capital is fixed at 100 units in the short run. Assume further that the price of capital is \$50, and the price of labor is \$15. If the market price of candy bars is \$2, how many units of labor should the firm hire in order to maximize its profit?

**GPT-4 RESPONSE**

To determine how many units of labor the firm should hire in order to maximize its profit, we first need to set up our problem and then use marginal productivity of labor (MP_L) in comparison to the wage rate (w).

Given:

$Q = 50K^{0.5} L^{0.5}$

$K = 100$

$r = \$50$ (price of capital)

$w = \$15$ (price of labor)

$P = \$2$ (price of candy bars)

1. Production Function with Given Capital:

Plugging in $K = 100$:

$Q = 50(100)^{0.5} L^{0.5}$

This simplifies to:

$Q = 500L^{0.5}$

2. Marginal Product of Labor (MP_L):

To determine the additional output generated by hiring an additional unit of labor, differentiate Q with respect to L:

$MP_L = \frac{dQ}{dL}$

Differentiating $500L^{0.5}$:

$MP_L = 250L^{(-0.5)}$

3. Profit Maximization Condition:

For profit maximization, a firm in a perfectly competitive market should set the wage rate (w) equal to the marginal revenue product of labor (MRP_L). The MRP_L is the additional revenue generated by hiring an additional unit of labor and is given by:

$MRP_L = P \times MP_L$

So for profit maximization:

$w = P \times MP_L$

Given $w = \$15$ and $P = \$2$:

$15 = 2 \times 250L^{(-0.5)}$

$15 = 500L^{(-0.5)}$

Divide both sides by 500:

$L^{-0.5} = \frac{15}{500} = 0.03$

Now, taking the reciprocal:

$L^{0.5} = \frac{1}{0.03}$

$L^{0.5} = 33.33$

Squaring both sides to solve for L:

$L = (33.33)^2$

$L = 1111.11$

Therefore, the firm should hire approximately 1111 units of labor to maximize its profit.

## Implications for practice

Although it is important to understand the capability and limitations of various AI chatbots that use generative large language models, such as ChatGPT, Bard, and LLaMA 2, it is perhaps even more useful to reflect on how learning, teaching, and assessment strategies need to be rethought in response to these new, powerful technologies.

For students, AI chatbots offer unprecedented opportunities to understand concepts and solve problems, whether in the field of economics or any other area of study. To give five concrete examples:

1. Students can ask an AI chatbot to explain any topic in economics, such as "opportunity cost," "negative externality," or "national income accounting." In response to these inquiries, the student will not only receive a definition but also examples and explanations. If not prompted for specific application examples, its responses will be limited to explanations of the concept.

2. Students can ask an AI chatbot to solve a numerical problem, such as finding the market equilibrium when given demand and supply equations. However, different chatbots will differ in their ability to solve this problem. For example, GPT-3.5 is unable to find the correct market equilibrium in this case, while GPT-4 easily produces the correct solution.

3. Students can ask an AI chatbot to analyze different government policies, such as the impact of a sales tax on soda or a pollution tax. Similar problems were used for the advanced prompts and once again GPT-4 performed better than the alternative models.

4. Students can ask an AI chatbot to recommend a government policy to solve a real-world issue, such as problems associated with diabetes, poaching of elephants, and high rates of inflation. In each of these cases, ChatGPT will suggest long lists or varied and comprehensive policy initiatives that provide students with knowledge and ideas. These ideas can then be used for further study (perhaps by refining the prompt given to the AI chatbot–prompt engineering).

5. Students can ask an AI chatbot to write their essays or term papers on any topic imaginable. By inputting a single prompt and then using the AI response to refine the question or ask for specific examples, the student can easily produce a "five-paragraph essay" at any level of sophistication (rewrite the essay as a four-year-old; rewrite the essay as a PhD student in economics).

Clearly, generative language models can help students learn and perform any assessment task, if they have access to a computer (or a smartphone). Of course, as we have seen in this study, AI chatbots are prone to give incorrect conclusions and faulty and incomplete explanations. Thus, any student who ventures to use AI technology for learning or assessment must be aware of the risks associated with receiving (and using) incorrect explanations and solutions.

Naturally, the ability of students to answer any question that economics educators have traditionally used for both formative and summative assessment purposes raises serious concerns among education professionals. Given the easy access to these technologies, educators must reflect on how they design assessment activities and assess student work. At the same time, educators must consider how to use the promising potential for student learning that AI chatbots offer. Below five suggestions for using an AI chatbot to enhance student learning are offered:

1. It can provide students with instant feedback when they work on problems. That is, students can ask questions on any topics they find challenging or confusing. In this way, the AI chatbot can act like a personal tutor to a student, "saving instructors' time and energy" [43, p.1]. In contrast to an instructor or a teaching assistant, the AI chatbot is effortlessly available at any time of the day (or night).

2. An AI chatbot can create examples and scenarios that allow students to learn a topic more deeply. For example, learning the concept of opportunity cost usually requires seeing it in

"action" in many varied situations. It is difficult for an instructor to provide more than a few such examples, but an AI chatbot can easily generate example after example, tirelessly.

3. An AI chatbot can help students generate ideas for an essay on a specific topic, as well as help students refine their own ideas on the topic. In addition, students can easily use an AI chatbot to refine their writing [44], remove grammatical errors, and make the presentation of their ideas clear. In economics, writing may be secondary to understanding and the ability to apply concepts to problems and issues in society.

4. An AI chatbot can provide concise just-in-time explanations and examples on any topic that students encounter when reading a text or working on a research project. Or, for that matter, any issue they encounter while watching/reading the news or engaging with social media.

5. An AI chatbot can create practice problems, quizzes, and other types of educational material that students can use to better learn economics concepts in a flipped classroom context [45]. The technology is thus perfect for taking advantage of the testing effect, the increase in long-term learning facilitated by retrieving information from memory [46].

More generally, AI technology is well-suited for applying the five evidence-based learning strategies suggested by cognitive load theory [47–49]. According to cognitive load theory, deep and lasting learning are promoted through (1) distributed practice (spacing study sessions over time), (2) interleaving (mixing related, but distinct material), (3) retrieval practice (retrieving information from memory), (4) elaboration (explaining material with many details to make connections), and (5) concrete examples [50].

## Limitations

While this study sheds light on the efficacy and potential use of AI chatbots in higher education, especially in economics, there are certain limitations that need acknowledgment. First, the focus of the study was primarily on introductory and intermediate economics. Whether the findings can be generalized to other academic disciplines is an aspect that has not been evaluated in this study. Second, AI chatbots are trained on vast amounts of data, which may contain biases. The potential risks associated with these biases or the ethical considerations of using AI for educational purposes were not investigated in this study. Further studies, which take these variables into account, will need to be undertaken.

## Future directions

Despite the promising results of the study, questions remain. First, further work is needed to confirm and validate our findings. Second, further investigations may explore the performance and accuracy of other chatbots such as Anthropic's Claude [https://claude.ai/], You.com [https://you.com/], or Perplexity [https://www.perplexity.ai/] on economics prompts. Finally, further studies are needed to investigate whether chatbots such as Baidu's Ernie (Chinese) or Jais (Arabic) could accurately answer discipline-specific creative prompts in those two languages.

## Supporting information

**S1 Appendix. Advanced prompts.**
(DOCX)

## Author Contributions

**Conceptualization:** Patrik T. Hultberg.

**Formal analysis:** Patrik T. Hultberg, Firuz Kamalov, Linda Smail.

**Writing – original draft:** Patrik T. Hultberg, David Santandreu Calonge, Firuz Kamalov, Linda Smail.

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
