## [Decision Letter · Decision Letter 0]

16 Oct 2023

PONE-D-23-29171Who understands economics? Let’s ask the chatbotsPLOS ONE

Dear Dr. Hultberg,

Thank you for submitting your manuscript to PLOS ONE. After careful consideration, we feel that it has merit but does not fully meet PLOS ONE’s publication criteria as it currently stands. Therefore, we invite you to submit a revised version of the manuscript that addresses the points raised during the review process.

We look forward to receiving your revised manuscript.

Kind regards,

Takayuki Mizuno, Ph. D.

Academic Editor

PLOS ONE

Journal Requirements:

Additional Editor Comments:

The three reviewers were divided in their evaluations for this paper. However, all three reviewers agreed that the research topic is interesting. The three reviewers have different levels of rigor regarding the results of the research that they require for acceptance of the paper. The author should take Reviewer 1's comments seriously and revise the paper to increase the rigor of the research results.

Reviewers' comments:

Reviewer's Responses to Questions

**Comments to the Author**

1. Is the manuscript technically sound, and do the data support the conclusions?

Reviewer #1: No

Reviewer #2: Yes

Reviewer #3: Yes

2. Has the statistical analysis been performed appropriately and rigorously? 

Reviewer #1: No

Reviewer #2: Yes

Reviewer #3: Yes

3. Have the authors made all data underlying the findings in their manuscript fully available?

Reviewer #1: Yes

Reviewer #2: Yes

Reviewer #3: Yes

4. Is the manuscript presented in an intelligible fashion and written in standard English?

Reviewer #1: Yes

Reviewer #2: Yes

Reviewer #3: No

5. Review Comments to the Author

Reviewer #1: In the contemporary academic landscape, the study under review garners attention due to its pertinent subject matter. This research critically evaluates the accuracy and explanatory quality of four cutting-edge AI chatbots, namely GPT-3.5, GPT-4, Bard, and LLaMA 2, within the framework of university-level economics discourse. While the premise of the research promises novelty and insight, several concerns about its execution and presentation arise.

To begin with, the paper, upon thorough examination, does not appear to offer any novel scientific results or notable contributions to the existing body of literature. Such a lacuna undermines its potential value to the academic community. Additionally, a fundamental tenet of empirical research is the formulation and testing of hypotheses. Regrettably, this paper neither proposes any clear hypotheses nor displays the requisite rigor in methodologies to either substantiate or refute them. This absence further exacerbates concerns regarding the study's scientific integrity.

Moreover, the presentation of data, particularly the quality of the charts, is unsatisfactory. Clear and high-resolution graphics are crucial for the concise conveyance of research findings, and the compromised quality here diminishes the user experience and comprehension.

Given the outlined deficiencies—namely, the lack of discernible scientific contribution, the absence of hypotheses and methodological rigor, and subpar graphical presentation—it is recommended that this paper be declined for publication.

Reviewer #2: Dear Authors,

Thank you for the opportunity to review your manuscript. I can see much to be positive about with this study, particularly with the timeliness and need for greater analysis of such research questions. The following comments are written with acknowledgement that this appears to be a reasonably sound study with an important contribution to the literature, and the comments made are with enhancing its possible rigor, transparency, and impact.

1. The study relies on the ability to translate human prompts (i.e., test questions) into statements that the AI can respond to. While it looks like the prompts used are equivalent to a typical exam question, can the authors consider a supplementary test where appropriate prompt engineering (see Eager & Brunton, 2023) is used in the design. While I wouldn't suggest replacing the current findings, i.e., that AI can complete traditional test formats satisfactorily, but also the degree to which this changes when effective prompt engineering is deployed in recrafting the test.

2. Table 5 and 6 could be formatted better, it seems much more confusing than it needs to be. Consider reviewing an APA7 t-test table, and using that as a guide.

3. Consistency on raters. Given this study relies heavily on the use of subjective assessment of the students/AI, there ought to be greater information provided regarding how inter-rater reliability was assured (e.g., Lievens, 2001). This might include information on if those in this study are the same academics that rated the ~2000 students and the AI, and appropriate re-sampling of those student-tests with the research team and the AI-tests to assess for consistency among judgments.

4. I can't make a PLOS-specific judgement on the discussion and conclusions, but for a higher education manuscript, this section seems insufficient in size and scope. I would have expected a stronger connection to how this study contributes to, conflicts with, or confirms previous studies.

5. The implications seem simplistic. Yes, it does say students can use AI, but we know this. What does this paper say about academic integrity and assessment design within economics? And what can economics professors do about it?

I look forward to re-reading your study.

References

Eager, B., & Brunton, R. (2023). Prompting higher education towards AI-augmented teaching and learning practice. Journal of University Teaching & Learning Practice, 20(5), 02.

Lievens, F. (2001). Assessor training strategies and their effects on accuracy, interrater reliability, and discriminant validity. Journal of Applied Psychology, 86(2), 255.

Reviewer #3: This is an interesting article assessing different large language models. The article is well-written. Authors may think revising the title for making it more suitable for scientific journal also think about SEO during writing the title.

6. PLOS authors have the option to publish the peer review history of their article (what does this mean?). If published, this will include your full peer review and any attached files.

Reviewer #1: **Yes: **Batyrkhan Omarov

Reviewer #2: **Yes: **Dr Joseph Crawford

Reviewer #3: No

---

## [Author Response · Author response to Decision Letter 0]

17 Nov 2023

Editor’s Comments 

Please ensure that your manuscript meets PLOS ONE's style requirements, including those for file naming 

Authors’ responses

We apologize and have uploaded all the revised files according to your specifications. In particular, the “Revised Manuscript with Track Changes” now includes a title page with all authors and affiliations included, in addition to all changes highlighted. A clean revised manuscript has also been uploaded. 

The references have been updated and revised and the data used in the study have been uploaded on Git Hub and can be accessed through the following link:

https://github.com/group-automorphism/tuce

Editor’s Comments 

Please ensure that you include a title page within your main document. Could you therefore please include the title page into the beginning of your manuscript file itself, listing all authors and affiliations. 

Authors’ responses

Yes, the title page has now been added to the “Revised Manuscript with Track Changes” file.

Editor’s Comments 

If there are no restrictions, please upload the minimal anonymized data set necessary to replicate your study findings as either Supporting Information files or to a stable, public repository and provide us with the relevant URLs, DOIs, or accession numbers.

Authors’ responses

There are no restrictions and the data used has been uploaded on Git Hub and can be accessed through the following link:

https://github.com/group-automorphism/tuce

Reviewer #1 

The paper, upon thorough examination, does not appear to offer any novel scientific results or notable contributions to the existing body of literature. 

Authors’ responses

We appreciate your thoughtful review of our research comparing four chatbots in the field of economics. Your concern regarding the novelty and notable contributions of our work is duly noted, and we would like to take this opportunity to address your points.

Research on comparing chatbots, as outlined in papers by Plevris, V., et al. (2023) and Calonge, D. S., et al. (2023) specifically focuses on assessing the capabilities of AI chatbots in solving math and logic problems, as well as their utility in the domains of calculus and statistics. Our research, however, compares for the first time in the published literature 4 (not 1, 2 or 3, latest released versions of) chatbots, namely GPT-3.5, GPT-4, Bard, and LLaMA 2, for accuracy of conclusions and quality of explanations in the context of university-level economics, specifically 60 questions from the Test of Understanding in College Economics (TUCE) + 10 advanced economics prompts. 

Although a similar exercise was conducted on a different version of TUCE using only GPT-3.5 (Geerling et al., 2023), the current research adds not only 3 additional chatbots but also includes a careful analysis of the explanations provided by the chatbots; that is, the research goes beyond simply assessing whether the response was correct or incorrect. 

A search on most used databases (Scholar, Scopus, etc.) would evidence this. And we believe that our work and the methodology used in our article offer several important contributions to the broader field of artificial intelligence and pedagogical technology, which can directly benefit economic research and education.

Methodological Framework: Our research provides a structured methodological framework for evaluating AI chatbots in a range of disciplines. This framework can be adopted and adapted by researchers in various fields, including economics, who wish to assess AI tools for their specific domain.

Comparative Analysis: We conduct a comprehensive comparison of 4 different chatbots, including their performance, response quality, and user experience. This comparison can serve as a benchmark for future studies that aim to evaluate chatbots in economic (or other disciplines) applications, enabling researchers to understand the state of the art in chatbot capabilities.

Pedagogical Potential: The use of chatbots in education is a growing field, and this work can inform educators, students, and researchers in economics about the potential of chatbots as teaching assistants. While we focus on economics problems, the insights gained can be relevant to other educators looking to employ chatbots as instructional tools.

Limited Existing Literature: As we pointed out, there is limited published literature on this specific topic in economics. By initiating this conversation, our research opens the door for further exploration in the intersection of AI chatbots and economics, potentially leading to more specialized studies.

Considering these contributions, we believe our research has value not only in its immediate domain but also in its potential to inform and stimulate further research in economics and related fields. 

Reviewer #1 

A fundamental tent of empirical research is the formulation and testing of hypotheses. Regrettably, this paper neither proposes any clear hypotheses nor displays the requisite rigor in methodologies to substantiate or refute them.

Authors’ responses

Thank you for this comment. In the original submission (lines 302-308) we included the following sentence:

While it is evident from Table 4 that there is a difference between the performances of the chatbots on TUCE III, the difference may not be statistically significant given the relatively small sample size. To ascertain the significance of the difference in accuracy, a two-tailed, two-sample test of proportions was conducted. The null hypothesis in the test is that the proportion of correct answers is the same for the pair of chatbots under consideration; that is, that the two chatbots have the same accuracy. (revised lines 347-353)

To further highlight our hypothesis, we also added the following:

OpenAI describes GPT-4 as an enhanced version of GPT-3.5 and has lauded its capabilities. To test this assertion, the current research hypothesizes that GPT-4 performs significantly better than GPT-3.5, Bard, and Llama 2 in the context of undergraduate economics. To test this hypothesis, each model was evaluated on TUCE III, as well as more advanced prompts in economics. The null hypothesis was thus that each AI chatbot would score the same across all economics prompts. The results, as shown in Tables 4, 5, and 6, reject the null hypothesis and indicated that the difference between GPT-4 and the other algorithms is statistically significant in the context of undergraduate economics. [lines 305-312]

The main hypothesis has also been added to the abstract:

The null hypothesis that all AI chatbots perform equally well on prompts that explore understanding of economics is rejected.

Reviewer #1

The presentation of data, particularly the quality of the charts, is unsatisfactory. Clear and high-resolution graphics are crucial for the concise conveyance of research findings, and the compromised quality here diminishes the user experience and comprehension 

Authors’ responses

All figures were remade and saved in a higher-resolution graphic. 

Figures 1 and 2 are similar to the figures submitted originally, with the exception for the higher resolution.

Figures 3 and 4 were re-configured to raise their readability and ease-of-access, and then saved at a higher resolution. 

Reviewer #2 

Can the authors consider a supplementary test where appropriate prompt engineering (see Eager & Brunton, 2023) is used in the design 

Authors’ responses

Thank you for bringing up the paper by Eager and Brunton (2023), and its relevance to our research. We appreciate the opportunity to clarify its relationship to our work and the specific focus of our study.

While Eager and Brunton's paper primarily addresses the importance of prompt engineering in guiding the generation of quality outputs from AI models, we agree that their focus is on instructional text in a general context, on “the usefulness of AI tools when generating teaching and learning content (p.1), rather than a specific discipline like economics. Our research’s focus was not on generate content but rather generate accurate responses to TUCE and our advanced prompts. Our research is indeed more closely aligned with the educational application of AI chatbots and their usefulness (for students and academics) in the field of economics.

In our study, we replicate the work by Geerling et al. (2023) related to the Test of Understanding in College Economics (TUCE). We extended this replication by utilizing a different set of questions, namely from a previous standardized test of College Economics (TUCE III). This extension is intended to provide a more comprehensive evaluation of the chatbots' capabilities and performance in the domain of economics.

Furthermore, we incorporated advanced (challenging) prompts, as detailed in Appendix 1 of our paper. These prompts were designed to rigorously test the chatbots using prompt engineering techniques, as advocated by Eager and Brunton. The prompts were constructed with the use of action verbs based on Bloom's Taxonomy, a widely accepted framework for creating learning outcomes (Biggs & Collins, 2014). In other words, the ten additional questions were indeed “engineered” by us, but not in an effort to optimize the responses provided by the chatbots, but rather to evaluate if the chatbots can respond accurately to more complex prompts. By using these advanced prompts, we aimed to assess the chatbots' ability to engage in higher-order thinking and provide nuanced responses, which is particularly relevant in an educational context.

In summary, while our research aligns with the principles of prompt engineering, we have applied them to evaluate chatbots in the specific domain of economics, building upon existing literature in this area. We believe that this approach contributes to the understanding of chatbots' utility in a specialized field and offers insights into their performance when confronted with more challenging and domain-specific prompts.

We appreciate your thorough review and hope this clarification underscores the significance of our research within the context of economics and AI chatbot assessment.

Reviewer #2

Table 5 and 6 could be formatted better, it seems much more confusing than it needs to be. Consider reviewing an APA7 t-test table and using that as a guide.

Authors’ responses

All tables were re-formatted in terms of font and spacing. In addition, tables that included t-test results were adjusted to reflect more standard presentation styles. Finally, all shading was removed to clean up the presentation. Thank you. 

Reviewer #2

Consistency on raters. Given this study relies heavily on the use of subjective assessment of the students/AI, there ought to be greater information provided regarding how inter-rater reliability was assured (e.g., Lievens, 2001). This might include information on if those in this study are the same academics that rated the ~2000 students and the AI, and appropriate re-sampling of those student-tests with the research team and the AI-tests to assess for consistency among judgments.

Authors’ responses

The authors of this article are not those who rated the 2000 students. More specifically, the 2,000 plus students that were evaluated using the Test of Understanding College Economics (TUCE) as described by Saunders (1991). The data was collected from a sample of schools during academic year 1989-90.

The authors of this article employed Krippendorff's alpha coefficient, as introduced by Krippendorff in 2011, to gauge the level of agreement and inter-rater reliability in our analysis of the responses generated by the four chatbots (0.87). The primary objective of employing this measure was to mitigate subjectivity in our analyses and to uphold consistency among different raters.

Krippendorff's alpha is a versatile measure that can be applied to assess inter-rater reliability in situations where there are multiple raters and variables, and it accommodates both nominal and ordinal data. Given the complexity of evaluating the subtleties and nuanced responses to economics prompts (TUCE and our advanced prompts) generated by the AI chatbots, particularly in the context of our research, Krippendorff's alpha offered a robust and adaptable method to assess the degree of agreement among raters and ensure that our findings were reliable and consistent.

[A paragraph was added to the Methods section to convey this information.]

Cohen's Kappa is another commonly used measure for inter-rater reliability, but we felt that it is more suitable for binary or nominal data and may not be as versatile as Krippendorff's alpha, which is better suited for a broader range of data types and complexity.

Krippendorff, K. (2011). Computing Krippendorff's alpha-reliability. Document retrieved from https://repository.upenn.edu/asc_papers/43.

Reviewer #3 

This is an interesting article assessing different large language models. The article is well-written. Authors may think revising the title for making it more suitable for scientific journal also think about SEO during writing the title.

Authors’ responses

Thank you for your kind comment. The title was revised to read:

Comparing and Assessing Four AI Chatbots' Competence in Economics

---

## [Editor Report · Decision Letter 1]

15 Jan 2024

Comparing and Assessing Four AI Chatbots' Competence in Economics

PONE-D-23-29171R1

Dear Dr. Hultberg,

We’re pleased to inform you that your manuscript has been judged scientifically suitable for publication and will be formally accepted for publication once it meets all outstanding technical requirements.

Kind regards,

Takayuki Mizuno, Ph. D.

Academic Editor

PLOS ONE

Additional Editor Comments (optional):

I have verified that the author has correctly incorporated all of the reviewer's comments into the paper.
---

## [Editor Report · Acceptance letter]

26 Apr 2024

PONE-D-23-29171R1 

PLOS ONE

Dear Dr. Hultberg, 

I'm pleased to inform you that your manuscript has been deemed suitable for publication in PLOS ONE. Congratulations! Your manuscript is now being handed over to our production team.

Kind regards, 

on behalf of

Dr. Takayuki Mizuno 

Academic Editor

PLOS ONE